# Fold2Seq: A Joint Sequence(1D)-Fold(3D) Embedding-based Generative Model for Protein Design

## Abstract

Designing novel protein sequences consistent with a desired 3D structure or fold, often referred to as the inverse protein folding problem, is a central, but non-trivial, task in protein engineering. It has a wide range of applications in energy, biomedicine, and materials science. However, challenges exist due to the complex sequence-fold relationship and difficulties associated with modeling 3D folds. To overcome these challenges, we propose **Fold2Seq**, a novel transformer-based generative framework for designing protein sequences conditioned on a specific fold. Our model learns a fold embedding from the density of the secondary structural elements in 3D voxels, and then models the complex sequence-structure relationship by learning a joint sequence-fold embedding. Experiments on high-resolution, complete, and single-structure test set demonstrate improved performance of **Fold2Seq** in terms of speed and reliability for sequence design, compared to existing baselines including the state-of-the-art RosettaDesign and other neural net-based approaches. The unique advantages of fold-based Fold2Seq becomes more evident on diverse real-world test sets comprised of low-resolution, incomplete, or ensemble structures, in comparison to a structure-based model.

## 1 Introduction

Computational protein design is the conceptual inverse of the protein structure prediction problem, and aims to infer an amino acid sequence that will fold into a given 3D structure. Designing protein sequences that will fold into a desired structure has a broad range of applications, from therapeutics to materials (Kraemer-Pecore et al., 2001). Despite significant advancements in methodologies as well as in computing power, inverse protein design still remains challenging, primarily due to the vast size of the sequence space - and the difficulty of learning a function that maps from the 3D structure space to the sequence space. Earlier works rely mostly on energy minimization-based approaches (Koga et al., 2012; Rocklin et al., 2017; Huang et al., 2011), which follow a scoring function (force fields, statistical potentials, or machine learning (ML) models,) and sample both sequence and conformational space. Such methods often suffer from drawbacks such as low accuracy of energy functions or force-fields (Khan & Vihinen, 2010) and low efficiency in sequence and conformational search (Koga et al., 2012).

Recently, as the data on both protein sequences (hundreds of millions) and structures (a few hundreds of thousands) is quickly accumulating, data-driven approaches for inverse protein design are rapidly emerging (Greener et al., 2018; Karimi et al., 2020; Ingraham et al., 2019). Generally, data-driven protein design, attempts to model the probability distribution over sequences conditioned on the structures: $P(x|y)$, where $x$ and $y$ are protein sequences and structures, respectively. Two key challenges remain: (1) defining a good representation ($y$) of the protein structure and (2) modelling the sequence generation process conditioned on $y$. Current protein design methods use protein backbone information from a single protein structure (fixed backbone) or from a set of backbone structures consistent with a single fold (flexible backbone). In earlier studies, the protein structures are represented as a 1D string (Greener et al., 2018), a 1D vector (Karimi et al., 2020), a 2D image (Strokach et al., 2020), or a graph (Ingraham et al., 2019). The sequence generation methods used in the protein design studies can be classified as non-autoregressive (Karimi et al., 2020; Greener et al., 2018; Strokach et al., 2020) and autoregressive (Ingraham et al., 2019; Madani et al., 2020).

In non-autoregerssive methods, $\boldsymbol{y}$ is usually concatenated with a Gaussian random noise $\boldsymbol{z}$ (which is the latent vector of the sequence) please check here to be the input to a sequence generator $P(\boldsymbol{x}|\boldsymbol{y}) = f_g(\boldsymbol{y}, \boldsymbol{z})$, while in autoregressive methods, $P(\boldsymbol{x}|\boldsymbol{y})$ is decomposed through the chain rule: $P(\boldsymbol{x}|\boldsymbol{y}) = \prod_{i=1}^{n} P(x_i|x_1, x_2, ..., x_{i-1}, \boldsymbol{y})$, where $\boldsymbol{x} = (x_1, x_2, ..., x_n)$.

In this paper, we focus on designing sequences conditioned on a protein fold. A protein fold is defined as the arrangement (or topology) of the secondary structure elements of the protein relative to each other (Hou et al., 2003). A secondary structural element can be defined as the three dimensional form of local segments of a protein sequence. Protein folds are therefore necessarily based on sets of structural elements that distinguish domains. As protein structure is inherently hierarchical, the complete native structure can have multiple folds and a fold can be present in many protein structures. A single structure (fixed backbone) or an ensemble of structures (flexible backbone) can be used as representatives of a fold. The ensemble representation is often a better choice, as it captures the protein dynamics.

Despite the recent progress in using ML models for protein design, significant gaps still remain in addressing both aforementioned challenges (fold representation and conditional sequence generation). **First**, the current fold representation methods are either hand-designed, or constrained and do not capture the complete original fold space (See Sec. 2.2 for details), resulting in low generalization capacity or efficiency. **Second**, the sequence encoding and the fold encoding are learned separately in previous methods, which makes two latent domains heterogeneous. Such heterogeneity across two domains actually increases the difficulty of learning the complex sequence–fold relationship.

To fill the aforementioned gaps, the **main contributions** of this work are as follows:

- We propose a novel fold representation, through first representing the 3D structure by the voxels of the density of secondary structures elements (SSEs), and then learning the fold representation through a transformer-based structure encoder. Compared to previous fold representations, this representation has several advantages: first, it preserves the entire spatial information of SSEs. Second, it does not need any pre-defined rules, so that the paramterized fold space is not neither limited or biased toward any particular fold. Third, the representation can be automatically extracted from a given protein structure. Lastly, the density model also loosens the rigidity of structures so that the structural variation and lack of structural information of the protein is better handled.

- We employ a novel joint sequence-fold embedding learning framework into the transformer-based auto-encoder model. By learning a joint latent space between sequences and folds, our model, **Fold2Seq**, mitigates the heterogeneity between two different domains and is able to better capture the sequence-fold relationship, as reflected in the results.

- Experiments on standard test sets demonstrate that Fold2Seq has superior performance on perplexity, native sequence recovery rate, and native structure recovery accuracy, when compared to competing methods including the state-of-the-art RosettaDesign and other neural net models. Ablation study shows that this superior performance is directly attributed to our algorithmic innovations. Experiments on real-world test sets further demonstrates the unique practical utility and versatility of Fold2Seq compared to the structure-based baselines.

## 2 RELATED WORK

**Data-driven Protein Design** A significant surge of protein design studies that deeply exploit the data through modern artificial intelligence algorithms has been witnessed in the last two years. Greener et al. (2018) used the conditional variational autoencoder for generating protein sequences conditioned on a given fold. Karimi et al. (2020) developed a guided conditional Wasserstein Generative Adversarial Networks (gcWGAN) also for inverse fold design. Madani et al. (2020) trained an extreme large (1.2B parameters) language model conditioned on taxonomic and keyword tags such as molecular functions for generating protein sequences. Ingraham et al. (2019) developed a graph-based transformer for generating protein sequences conditioned on a fixed backbone. Lastly, Strokach et al. (2020) formulated the inverse protein design as a constraint satisfaction problem (CSP) and applied the graph neural networks for generating protein sequences conditioned on the residue-residue distance map that is a static representation of the structure.

**Protein Fold Representation** For an extensive overview of molecular representations, including those of proteins, please see David et al. (2020). Murzin et al. (1995) and Orengo et al. (1997) manually classified protein structures in a hierarchical manner based on their structural similarity. These classifications can be regarded as one-hot encoding of the fold representations. Taylor (2002) represents a protein fold using a "periodic table". This representation was later used for inverse fold design (Greener et al., 2018). However, it only considers three pre-defined folds (( layer, layer and barrel) for a set of structures, which significantly limits the spatial information content of the fold. Hou et al. (2003) chose hundreds of representative proteins and calculated the similarity scores among them. This similarity matrix was then used for kernel Principle Component Analysis (kPCA). A similar idea was used in Karimi et al. (2020) for inverse protein design. This representation needs a pre-defined set ((alpha only, beta only, and alpha+beta) of structures along with a similarity metric. Such representation could lead to biased or constrained representations of the fold space and also may not preserve the detailed spatial information of the fold. Finally, Koga et al. (2012) summarized three rules that describe the junctions between adjacent secondary structure elements for a specific fold. Again, these rules are hand designed for a subset of structures, which makes the representation restricted to a small part of the fold space and offers limited generalizability during conditional sequence generation.

**Joint Embedding Learning** Joint embedding learning across multiple different data modalities was first proposed by Ngiam et al. (2011) on audio and video signals. Since then, such approaches have been then widely used in cross modal retrieval or captioning tasks (Arandjelovic & Zisserman, 2018; Gu et al., 2018; Peng & Qi, 2019; Chen et al., 2018; Wang et al., 2013; Dognin et al., 2019). In few/zero-shot learning, joint feature-label embedding was used for predicting the label of instances belonging to unseen classes (Zhang & Saligrama, 2016; Socher et al., 2013). Several papers have demonstrated that learning joint embedding is useful for the single modal classification tasks (Ngiam et al., 2011; Wang et al., 2018; Toutanova et al., 2015). Moreover, Chen et al. (2018) used joint embedding learning for text to shape generation. Lastly, Joint sequence-label embedding is also applied for molecular prediction/generation (Cao & Shen, 2020; Das et al., 2018).

## 3 METHODS

### 3.1 BACKGROUND AND NOTATION

A protein is formed by a linear chain of amino acids (residues) that defines its 1D sequence. Chemical nature, as well as physical and chemical interactions with neighboring residues drive the folding of a sequence into different secondary structure elements or SSEs (helix, beta-sheet, loop, etc., see Fig 1(a)), that eventually forms a complete native 3D structure. A **protein fold** captures the topology and the composition of secondary structure elements, thus serving as an intermediate between the 1D sequence and the full 3D structure.

### 3.2 STRUCTURE REPRESENTATION THROUGH 3D VOXELS OF THE DENSITY OF SSES

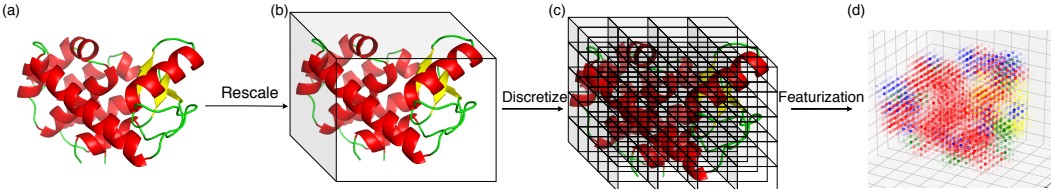

Figure 1: (a) The structure of PDB 107L. The secondary structures are colored as: helices in red, beta sheets in yellow and loops in green. (b) The structure is rescaled to fit the $40\text{Å} \times 40\text{Å} \times 40\text{Å}$ cubic box. (c) The box is discretized into voxels. (d) Features of each voxel are obtained from the structure content of the voxel (bend/turn in blue).

In this subsection, we describe how we represent the 3D structure to explicitly capture the fold information. We denote the position(3D coordinate) of each residue by its alpha carbon. For a given protein with length $N$, we first translate the structure to match its center of mass with the origin of the coordinate system. We then rotate the protein around the origin to let the first residue be on the

negative side of z-axis. We denote the resulting residue coordinates as $c_1, c_2, ..., c_N$. We assign the secondary structure label to each residue based on their SSE assignment (Kabsch & Sander, 1983) in Protein Data Bank (Berman et al., 2000). We consider 4 types of secondary structure labels: helix, beta strand, loop and bend/turn. In order to consider the distribution of different secondary structure labels in the 3D space, we discretize the 3D space into voxels, as shown in Fig 1. A technical challenge here is that the sizes of different proteins vary drastically. As we are only considering the arrangement of SSEs, not their exact coordinates, we here rescale the original structure, so that it fits into a fixed-size cubic box. Based on the distribution of sizes of single-chain proteins in the CATH database (Sillitoe et al., 2019), we choose a $40\text{Å} \times 40\text{Å} \times 40\text{Å}$ box with each voxel of size $2\text{Å} \times 2\text{Å} \times 2\text{Å}$. We denote the scaling ratio as $r \in \mathcal{R}^3$. For voxel $i$, we denote the coordinates of its center as $v_i$. We assume that the contribution of residues $j$ to voxel $i$ follows the Gaussian form:

$$y_{ij} = \exp(-\frac{||c_j \odot r - v_i||_2^2}{\sigma^2}) \cdot t_j, \tag{1}$$

where $t_j \in \{0, 1\}^4$ is the one-hot encoding of the secondary structure label of amino acid $j$. The standard deviation is chosen to be $2\text{Å}$. We sum up all residues together to obtain the final features of the voxel $i$: $y_i = \sum_{j=1}^N y_{ij}$. The structure representation $y$ is the vector of $y_i$'s over all voxels. In the next subsection, we will describe how we encode $y$ into the latent space and how we learn a joint sequence-fold embedding in order for generating sequences consistent with a desired 3D structure.

### 3.3 Fold2Seq with Joint Sequence-Fold Embedding

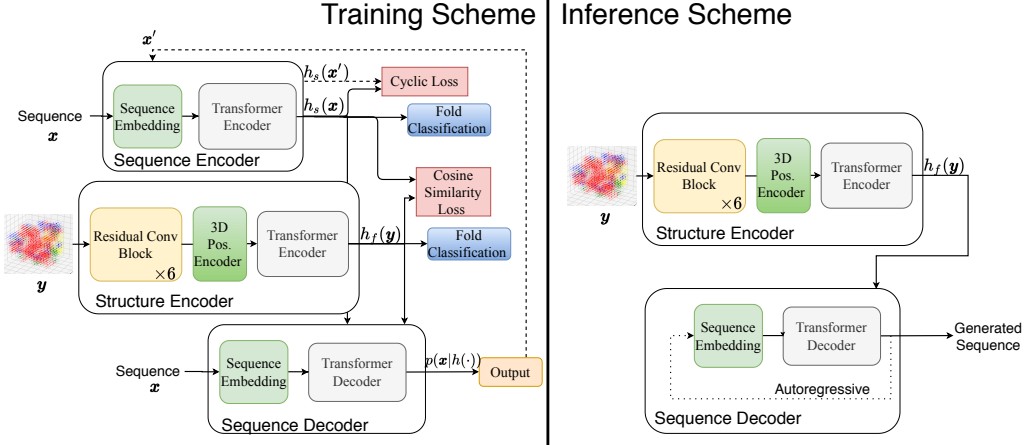

Figure 2: The architecture of the model during the training and inference stages. (**Training Scheme**): During training, the model includes three major components: (top) Sequence Encoder, (middle) Structure Encoder and (bottom) Sequence Decoder. The dashed arrows represent the process for getting cyclic loss. (**Inference Scheme**): During the inference, the model only needs the structure encoder and the sequence decoder for decoding sequences. check here and the figure

**Model Architecture** In the training stage, our model consists of three major components: a sequence encoder: $h_s(\cdot)$, a structure encoder: $h_f(\cdot)$ and a sequence decoder: $p(x|h(\cdot))$, as shown in Fig. 2 (Left). Both sequence encoder and decoder are implemented using the vanilla transformer model (Vaswani et al., 2017). The structure encoder contains 6 residual blocks followed by a 3D positional encoding. Each residual block has two 3D-convolutional layers ($3 \times 3 \times 3$) and batch normalization layers. The 3D positional encoding is a simple 3D extension of the sinusoidal encoding described in the vanilla transformer model, as shown in Appendix A. After the positional encoding, the 3D vector is flattened to be 1D as the input of a transformer encoder. The length of the transformer input is fixed to be $l_f = 5^3 = 125$. The output of the transformer encoder: $h_f(y)$ is the latent fold representation of $y$.

We propose a simple fold-to-sequence reconstruction loss based on the auto-encoder model: $\mathbf{RE}_f = p(x|h_f(y))$. However, as mentioned earlier, training based on $\mathbf{RE}_f$ alone suffers due to the heterogeneity of $x$ and $y$. To overcome this challenge, we first encode the sequence $x$ through the

sequence encoder into the latent space as: $h_s(\boldsymbol{x})$, which could be done through a simple sequence-to-sequence reconstruction loss: $\mathbf{RE}_s = p(\boldsymbol{x}|h_s(\boldsymbol{x}))$. We then learn a joint latent space between $h_f(\boldsymbol{y})$ and $h_s(\boldsymbol{x})$ through a novel sequence-fold embedding learning framework.

**Joint Embedding Learning**  Typically, learning a joint embedding across two domains needs two intra-domain losses and one cross-domain loss (Chen et al., 2018). An intra-domain loss forces two semantically similar samples from the same domain to be close to each other in the latent space, while a cross-domain loss forces two semantically similar samples in different domains to be closer. In our case, the meaning of 'semantically similar' is that the proteins should have the same fold(s). Therefore, we consider a supervised learning task for learning intra-domain similarity: fold classification. Specifically, the outputs of both encoders: $h_f(\boldsymbol{y}) \in \mathcal{R}^{l_f \times d}$ and $h_s(\boldsymbol{x}) \in \mathcal{R}^{l_s \times d}$ will be averaged along $l_f$ and $l_s$ dimensions, followed by a MLP+softmax layer to perform fold classification (shown as two blue blocks in Fig. 2), where $l_s$, $l_f$ and $d$ are the length of the sequence, the fold and the latent state, respectively. The two MLP layers' parameters are shared. The category labels follow the fold (topology) level of hierarchical protein structure classification in CATH4.2 dataset (Sillitoe et al., 2019) (see Section 3.4). As a result, we propose the following two intra-domain losses: $\mathbf{FC}_f$ and $\mathbf{FC}_s$, the cross entropy losses of fold classification from $h_f(\boldsymbol{y})$ and $h_s(\boldsymbol{x})$ respectively. The benefits of these two classification tasks are obvious: First, it will force the structure encoder to learn the fold representation. Second, as we perform the same supervised learning task on the latent vectors from two domains, it will not only learn the intra-domain similarity, but also cross-domain similarity. However, without explicit cross-domain learning, the two latent vectors $h_f(\boldsymbol{y})$, $h_s(\boldsymbol{x})$ could still have minimal alignment between them. We will describe the cross-domain loss in the rest of this subsection.

In the transformer decoder, each element in the **non**-self attention matrix is calculated by the cosine similarity between the latent vectors from the encoder and the decoder, respectively. Inspired by this observation, we consider to maximize the cosine similarity (shown as the 'Cosine Similarity' pink block in Fig 2) between $h_f(\boldsymbol{y}) \in \mathcal{R}^{l_f \times d}$ and $h_s(\boldsymbol{x}) \in \mathcal{R}^{l_s \times d}$ as the cross-domain loss. We first calculate the matrix-product between $h_f(\boldsymbol{y})$ and $h_s(\boldsymbol{x})$ as $\boldsymbol{Q} = h_f(\boldsymbol{y}) \cdot h_s(\boldsymbol{x})^T, \boldsymbol{Q} \in \mathcal{R}^{l_f \times l_s}$. The $i$th row in $\boldsymbol{Q}$ represents the similarity between $i$th position in the fold and every position of the sequence. We would like to find the best-matching sequence piece with each position in the fold. To achieve this, the similarity matrix $\boldsymbol{Q}$ first goes through a row-wise average pooling with kernel size $k$, followed by the row-wise $\max$ operation:

$$\boldsymbol{q} = \max_{row}(\text{AvgPool}_{row}^k(\boldsymbol{Q})), \boldsymbol{q} \in \mathcal{R}^{l_f \times 1} \tag{2}$$

where $row$ means the operation is row-wised. We choose $k = 3$, which means the scores of every 3 continuous positions in the sequence will be averaged. We finally average over all positions in the fold to get the final similarity score: $\mathbf{CS} = \text{mean}(\boldsymbol{q})$. please read this subsection until here. I did some equation-in-line here.

Besides the cosine similarity loss, inspired from the earlier CycleGAN work (Zhu et al., 2017), we add a cyclic loss (shown as the 'Cyclic Loss' red block in Fig 2.) to be another term of our cross-domain loss. Specifically, we take the argmax of the output of fold-to-sequence model: $\boldsymbol{x}' = \arg\max p(\boldsymbol{x}|h_f(\boldsymbol{y}))$, and send it back to the sequence encoder for generating the cyclic-seq latent state: $h_s(\boldsymbol{x}')$ (shown as the dashed line in Fig 2). This cyclic-seq latent state will compare with the native seq latent state $h_s(\boldsymbol{x})$ through the square of the L2 distance:

$$\mathbf{CY} = ||h_s(\boldsymbol{x}') - h_s(\boldsymbol{x})||_2^2 \tag{3}$$

To summarize, the complete loss objective is the following:

$$L = \lambda_1 \mathbf{RE}_f + \lambda_2 \mathbf{RE}_s + \lambda_3 \mathbf{FC}_f + \lambda_4 \mathbf{FC}_s + \lambda_5(\mathbf{CY} - \mathbf{CS}) \tag{4}$$

where $\lambda_1$ through $\lambda_5$ are the hyperparameters for controlling the importance of these losses.

**Training and Decoding Strategy**  During experiments we found that, if the sequence encoder and the structure encoder were trained together, the structure encoder had little parameter improvement while the sequence encoder dominated the training. To overcome this issue, we consider a two-stage training strategy. In the first stage, we train the sequence-to-sequence model regularized by the sequence intra-domain loss: $L_1 = \lambda_2 \mathbf{RE}_s + \lambda_4 \mathbf{FC}_s$. After the first stage is finished, we start the

second training stage. We train the fold-to-sequence model regularized by the fold intra-domain loss and the cross-domain loss while keeping the sequence encoder frozen: $L_2 = \lambda_1 \mathbf{RE}_f + \lambda_3 \mathbf{FC}_f + \lambda_5(\mathbf{CY} - \mathbf{CS})$. The comparison between the one-stage training and two-stage training strategies are described in details in Appendix E.

We implement our model in Pytorch (Paszke et al., 2019). Each transformer block has 4 layers and $d = 256$ latent dimensions. In order to increase the robustness of our model for rotated structures, we augment our training data by right-hand rotating the each structure by $90°$, $180°$ and $270°$ along each axis (x,y,z). As a result, we augment our training data by $3 \times 3 = 9$ times. The learning rate schedule follows the original transformer paper (Vaswani et al., 2017). We use the exponential decay (Blundell et al., 2015) for $\lambda_5 = 1/2^{\#\text{epoch} - e}$ in the loss function, while $\lambda_1$ through $\lambda_4$ and $e$ are tuned based on the validation set, resulting $\lambda_1 = 1.0, \lambda_2 = 1.0, \lambda_3 = 0.02, \lambda_4 = 1.0, e = 3$. We train our model on 2 Tesla K80 GPUs, with batch size 128. In every training stage we train up to 200 epochs with an early stopping strategy based on the loss on the validation set. [1]

During inference, we only need the Fold2seq model for decoding sequences (Fig 2 (Right)). Top-k sampling strategy (Fan et al., 2018) is used for sequence generation, where $k$ is set to be 5 (tuned based on the validation set).

## 3.4 BENCHMARK DATASETS

We used the protein structure data from CATH 4.2 (Sillitoe et al., 2019) filtered by 100% sequence identity. We remove proteins that (1) are multi-chain or non-continuous in sequence; (2) contain other than 20 natural amino acids; (3) have length longer than 200. We randomly split the dataset based on the fold-level classification of protein structures into 95%, 2.5%, 2.5% as dataset (a), (b) and (c), which means that the three datasets have non-overlapping folds. We further randomly split the dataset (a) into 95%, 2.5% and 2.5% as dataset (a1), (a2) and (a3). Datasets (a1) (a2) (a3) have overlapping folds. We use dataset (a1) as the training set, (b)+(a2) as the validation set, (a3) as the In-distribution (ID) test set and (c) as the Out-of-distribution (OD) test set. The folds of the ID test set overlaps with the training set, whereas the folds of OD test set do not overlap with the training set. The statistics of these datasets are presented in Appendix B.

In order to quantitatively measure their difficulty levels, we calculate the averaged maximum sequence similarity (**amsi**) between a given test set $\boldsymbol{T}$ and the training set, defined as: $\text{amsi}_{\boldsymbol{T}} = \frac{1}{|D_{\boldsymbol{T}}|} \sum_{i \in D_{\boldsymbol{T}}} \max_{j \in D_{\text{train}}}(\text{SIM}(\boldsymbol{x}_i, \boldsymbol{x}_j))$, where $D_{\text{train}}$ and $D_{\boldsymbol{T}}$ are the training set and test set $\boldsymbol{T}$, respectively; $\text{SIM}(\boldsymbol{x}_i, \boldsymbol{x}_j)^2$ is the sequence similarity between sequence $\boldsymbol{x}_i$ and $\boldsymbol{x}_j$. As a result, we found we have $\text{amsi}_{\text{ID}} = 36.3\%$ and $\text{amsi}_{\text{OD}} = 16.3\%$. This shows that the sequence similarity between the ID test set and the training set is more than twice that between the OD test set and the training set. These values clearly demonstrate that the OD test set represents a much more difficult generalization task compared to the ID test set.

## 4 EXPERIMENTS ON BENCHMARK TEST SETS.

Ideally, the most appropriate criteria for evaluating inverse design methods is the extent that the structure of the generated sequence matches the desired fold/structure. However, as protein structure prediction is very time-consuming, and similar sequences usually indicate similar folds/structures, many earlier methods report performance in the sequence domain. We hereby perform comprehensive evaluations of Fold2Seq against two data-driven fold design methods: cVAE (Greener et al., 2018) and gcWGAN (Karimi et al., 2020) as well as the state-of-the-art principle-driven method, RosettaDesign[3] (Huang et al., 2011). For this comparison, we leave out methods that do not focus on inverse folding problem with a flexible backbone constraint (e.g. (Madani et al., 2020), (Strokach et al., 2020)). Three evaluation metrics are used: 1) Perplexity: Ability to assign high likelihood to

---

[1]We plan to release the code upon acceptance.

[2]Sequence similarity is measured through the Needleman Wunsch algorithm (Needleman & Wunsch, 1970) with Blossum62 scoring matrix.

[3]RosettaDesign is a principle-based method. It uses MCMC sampling and energy calculation to search for best sequences. The input to RosettaDesign consists of the backbone of the native structure and the SSE of each residue.

Table 1: Performance of different methods assessed in the sequence domain.

(a) Per-residue Perplexity.

| Model | ID Test | OD Test |
|---|---|---|
| Uniform | 20.0 | 20.0 |
| Natural | 18.0 | 18.0 |
| cVAE | 14.8 | 16.3 |
| gcWGAN | 13.5 | 15.2 |
| Fold2Seq | **8.1** | **11.9** |

(b) Avg. SeRR $\pm$ std. dev. (%).

| Model | ID Test | OD Test |
|---|---|---|
| Random across two folds | $12.8 \pm 7.94$ | $12.8 \pm 7.94$ |
| cVAE | $17.7 \pm 7.34$ | $15.3 \pm 5.34$ |
| gcWGAN | $17.5 \pm 6.35$ | $14.1 \pm 3.45$ |
| RosettaDesign | $20.3 \pm 5.13$ | $20.2 \pm 2.98$ |
| Fold2Seq | $\mathbf{27.1 \pm 6.31}$ | $\mathbf{24.1 \pm 2.64}$ |
| Random within same fold | $39.1 \pm 9.35$ | $39.1 \pm 9.35$ |

native (ground-truth) sequences; 2) Native sequence recovery rate (SeRR): Ability to recover native sequences; 3) Native structural recovery accuracy (StRA): Ability to recover native structures.

We note that the Graph_trans (Ingraham et al., 2019) model can be applied to a flexible backbone scenario. However, Graph_trans focuses on structure-based sequence design using the detailed backbone structure as an input, whereas the fold-based Fold2Seq designs sequences using the high-level fold information. Therefore, direct comparison between Fold2Seq and Graph_trans on the same train/test data containing full structure information is not fair. To highlight that Fold2Seq better captures fold information, we first re-train Graph_trans on our training set. We use tSNE to visualize the fold embeddings $h$ after the fold encoder for the proteins in the OD test sets. The results are shown in Appendix F. It is evident that the embeddings of same-fold proteins from Graph_trans are less clustered than those from Fold2Seq. Later, we will compare Fold2Seq with Graph_trans on three real-world design tasks, which will be described in details in Sec. 5.

**Perplexity and Sequence Recovery Comparison.** We first compare Fold2Seq against gcWGAN and cVAE in terms of per-residue Perplexity. Perplexity is a metric widely used in the language modeling and is not applicable to RosettaDesign. For reference, we also show the per-residue perplexity under the uniform distribution and the frequencies through all natural sequences in UniRef50 (Suzek et al., 2015). Performance on two test sets is summarized in Table 3a, showing that Fold2Seq has the smallest per-residue perplexity for both ID and OD test sets. We also note that the performance of all methods on the OD test set is worse than that on the ID test set, which matches our expectation of their difficulty levels.

Next, we compare the ability of different methods for recovering the native sequences given a desired fold. We calculate the averaged sequence similarity between the native sequence and the generated sequence on two test sets. Also, for comparison, we calculate the expected similarity between two random sequences in our whole dataset belonging to two different folds and belonging to the same folds. The results are summarized in Table 1b. In general, Fold2Seq performs significantly better, when compared to existing methods. Specifically, compared to RosettaDesign (second-best), Fold2Seq improves the recovery rate by 33.5% and by 19.3% on the ID and OD test set, respectively.

**Structural Recovery Comparison with RosettaDesign.** Although the comparison in sequence domain reflects the reliability of generated sequences to some extent, our final goal is to ensure that the structure of a generated sequence is similar to the native structure. Specifically, we used the iTasser Suite (Yang et al., 2015), one of the state-of-the-art protein structure prediction software, to predict the structure of the designed sequences. As iTasser usually takes at least one day for predicting the structure of a single protein, we only compare Fold2Seq to RosettaDesign in terms of the structure recovery. The similarity between two 3D protein structures is measured by the TM-score $\in [0, 1]$ (Zhang & Skolnick, 2004), where a higher score implies greater similarity.

For the $i$th protein in the test set, we compare $\text{TM}(s_i^{\textbf{Fold2Seq}}, s_i^{\textbf{Native}})$ against $\text{TM}(s_i^{\textbf{Rosetta}}, s_i^{\textbf{Native}})$, where $s_i^{\textbf{Fold2Seq}}$, $s_i^{\textbf{Rosetta}}$ and $s_i^{\textbf{Native}}$ are the 3D structures of the generated sequence from Fold2Seq, the generated sequence from RosettaDesign, and the native sequence, respectively.

We first compare the distributions of $\text{TM}(s_i^{\textbf{Fold2Seq}}, s_i^{\textbf{Native}})$ and $\text{TM}(s_i^{\textbf{Rosetta}}, s_i^{\textbf{Native}})$ on two test sets (ID and OD). The results are shown in Fig 3(a). For the ID test set, Fold2Seq shows significant improvement against RosettaDesign, while for the OD test set, Fold2Seq shows slightly better performance. We then define $\Delta\text{TM}_i = \text{TM}(s_i^{\textbf{Fold2Seq}}, s_i^{\textbf{Native}}) - \text{TM}(s_i^{\textbf{Rosetta}}, s_i^{\textbf{Native}})$ and perform

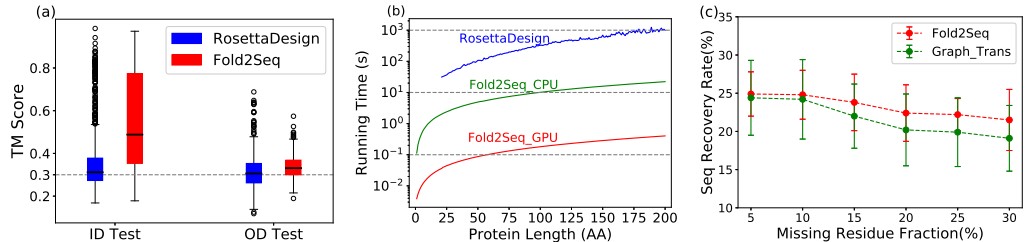

Figure 3: (a). TM score distributions of RosettaDesign and Fold2Seq. (b). Run time of Fold2Seq and RosettaDesign for generating one protein sequence: CPU: Intel Xeon E5-2680 v4 2.40GHz, GPU: Nvidia Tesla K80. (c). Avg. SeRR for the OD test set with a string of missing residues.

one-sided one-sampled t-test over $\Delta$TM, with null hypothesis as "$\Delta$TM $\leq 0.0$" on two test sets, revealing **P-value**$_{ID}$ $= 2.14E - 60$, **P-value**$_{OD}$ $= 0.019$, which demonstrate that Fold2Seq can overall generate more reliable structures compared to RosettaDesign. The two distributions over $\Delta$TM are shown in Fig 4 in Appendix C. We also randomly pick some designed structures with $\Delta$TM $> 0.0$ and $\Delta$TM $< 0.0$, and visualize them on Fig 5 and Fig 6 in Appendix D, respectively.

The sequence generation efficiency of Fold2Seq against RosettaDesign is shown in Fig 3(b). Fold2Seq is almost 100 times faster than RosettaDesign when running on CPU, and is 5000 times faster when running on GPU, while RosettaDesign can only run on CPU.

**Ablation Study.** In order to rigorously delineate the contributions of each algorithmic innovations, we perform an ablation study as following:

- cVAE: We use cVAE (Greener et al., 2018) as baseline with 1D string fold representation and MLP-based VAE.

- Trans_string_**RE**$_f$: We replace the MLP-based VAE in cVAE with transformer autoencoder model. The loss is $L = \mathbf{RE}_f$.

- Trans_voxel_**RE**$_f$: We replace the 1D string fold representation in "Trans_string_**RE**$_f$" with 3D voxel representation. We also add the convolutional residual block and 3D positional encoding. The loss is $L = \mathbf{RE}_f$.

- +**RE**$_s$+**CS**: We add the sequence encoder, together with the reconstruction loss and the cosine similarity loss to the previous loss: $L = \lambda_1 \mathbf{RE}_f + \lambda_2 \mathbf{RE}_s - \lambda_5 \mathbf{CS}$.

- +2**FC**: We add the two **FC** losses. $L = \lambda_1 \mathbf{RE}_f + \lambda_2 \mathbf{RE}_s + \lambda_3 \mathbf{FC}_f + \lambda_4 \mathbf{FC}_s - \lambda_5 \mathbf{CS}$.

- +**CY** (Fold2Seq): We add the cyclic loss into the former model with the final loss $L = \lambda_1 \mathbf{RE}_f + \lambda_2 \mathbf{RE}_s + \lambda_3 \mathbf{FC}_f + \lambda_4 \mathbf{FC}_s + \lambda_5 (\mathbf{CY} - \mathbf{CS})$.

We use average of native sequence recovery rate (SeRR) as the assessment metric. The performance on two test sets is summarized in Table 2a. Our **key** observations are: (i) By changing 'string' to 'voxel' and adding 2 **FC** losses, the performance has the largest improvement by 3-4%. (ii) By adding the cyclic loss (**CY**), the performance also improves around 2%. (iii) In contrast, by adding **RE**$_s$ and **CS**, the improvement is trivial. It also shows that including the two **FC** losses as the intra-domain loss is crucial for the joint embedding learning. In conclusion, our novel design of the 3D voxel representation and the joint embedding learning framework, which includes intra-domain and cyclic losses, results in significant improvement of the performance.

## 5 EXPERIMENTS ON REAL-WORLD DESIGN TASKS.

In order to further explore the practical utility of our model, we perform three real-world challenging design tasks: (1) Low-resolution structures; (2) Structure with region of missing residues; and (3) NMR ensembles. For (1) and (2), we compare with Graph_trans with a flexible backbone constraint which is re-trained on our training set.

**Low-Resolution Structures.** We first create the low-resolution structure dataset from Protein Data Bank, which contains 164 low resolution proteins (single-chain), resolution varying from 6Å to

Table 2: Average sequence recovery rate ±std. dev. (%) for (a) different models in ablation study, (b) low resolution structures and NMR ensemble.

(a)

| Model | ID Test | OD Test |
|---|---|---|
| cVAE | $17.7 \pm 7.34$ | $15.3 \pm 5.34$ |
| Trans_string_$\mathbf{RE}_f$ | $18.5 \pm 8.26$ | $16.1 \pm 3.23$ |
| Trans_voxel_$\mathbf{RE}_f$ | $22.3 \pm 8.23$ | $19.2 \pm 3.46$ |
| +$\mathbf{RE}_s$+$\mathbf{CS}$ | $22.4 \pm 7.54$ | $19.1 \pm 2.24$ |
| +$\mathbf{2FC}$ | $25.3 \pm 6.24$ | $22.2 \pm 2.13$ |
| +$\mathbf{CY}$ (Fold2Seq) | $\mathbf{27.1 \pm 6.31}$ | $\mathbf{24.1 \pm 2.64}$ |

(b)

| Model | Low_res Set | |
|---|---|---|
| Graph_trans | $19.9 \pm 4.8$ | |
| RosettaDesign | $17.2 \pm 6.3$ | |
| Fold2Seq | $\mathbf{21.2 \pm 3.1}$ | |
| Input | ID | OD |
| Single | $22.7 \pm 3.4$ | $20.9 \pm 4.2$ |
| Ensemble | $24.1 \pm 4.6$ | $22.3 \pm 3.1$ |

12Å. This set has maximum sequence similarity (MSI) below 30% to the training set. We compare Fold2Seq's performance on this set with that of Graph_trans and RosettaDesign. As shown in Table 2b, Fold2seq outperforms other baselines, because Fold2Seq only learns from the fold information by re-scaling the structure, discretizing the space, and smoothing the spatial secondary structure element information by neighborhood averaging. As Fold2Seq uses the high-level fold information, the method's performance is expected to be less sensitive compared to RosettaDesign or Graph_trans, when test structures are of lower resolution.

**Structures with missing string of residues.** We then perform the design task where the input structures have missing residues. In order to mimic the real-world scenario, for every protein in our OD test set, we select a stretch of residues at random starting positions with length $p$, for which residue information was removed. We compared Fold2Seq with Graph_trans at $p = \{5\%, 10\%, 15\%, 20\%, 25\%, 30\%\}$. As shown in Fig. 3c, initially, when $p$ is small, the performance of Fold2Seq is on par with Graph_trans. As $p$ increases, Fold2Seq outperforms Graph_trans with a consistent margin. This pattern matches our expectation, as Fold2seq is less sensitive to the availability of complete and detailed backbone structure information.

**NMR Structural Ensemble.** We finally apply Fold2Seq to a structural ensemble on NMR structures. We filter the NMR structures from our two test sets and obtain 57 proteins from the ID set and 10 proteins from the OD set. On average each protein has around 20 structures. Handling NMR ensembles using Fold2Seq is straightforward, when compared to Graph_trans and RosettaDesign: after we obtain the voxel-based features through Eq 1 for each model (structure) within one NMR ensemble, we simply average them across all models. Fold2Seq results for NMR ensembles are shown in Table 2b, along with a single structure baseline. Results show that Fold2Seq performs better on both ID and OD proteins, when ensemble structure information is available. This is consistent with the fold-based learning framework of Fold2Seq - as fold representation better captures structural variations present within a single fold.

## 6 Conclusion

In this paper, we design novel neural network model to learn a fold representation from the 3D voxels of density of secondary structure elements. In order to mitigate the heterogeneity between the sequence domain and the fold domain, we learn the joint sequence-fold representation through novel intra–domain and cross–domain losses. Our model, Fold2Seq outperforms existing data-driven methods and the state-of-the-art principle-driven method RosettaDesign, in terms of perplexity, sequence recovery rate, and structural recovery accuracy. We also show that our method is significantly faster compared to RosettaDesign. Ablation study shows that this superior performance can be directly attributed to our novel algorithmic innovations, including the fold representation, joint sequence-fold embedding, and various losses. Moreover, we demonstrate the unique practical utility of Fold2Seq compared to structure-based models in a set of real-world design tasks, including low resolution structures, structures with region of missing residues, and NMR structural ensembles.

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

## A 3D EXTENSION OF THE SINUSOIDAL ENCODING

We use a simple extension of the sinusoidal encoding described in the original transformer model (Vaswani et al., 2017) to encode each position in our Structure Encoder.

$$
\begin{aligned}
\text{PE}(x, y, z, 2i) &= \sin(x/10000^{2i/h}) + \sin(y/10000^{2i/h}) + \sin(z/10000^{2i/h}) \\
\text{PE}(x, y, z, 2i+1) &= \cos(x/10000^{2i/h}) + \cos(y/10000^{2i/h}) + \cos(z/10000^{2i/h})
\end{aligned}
\tag{5}
$$

## B DATASET STATISTICS

The statistics of our various datasets are given below.

- Training set includes 45995 proteins belonging to total 1093 folds.
- Validation set includes 1590 proteins belonging to total 256 folds.
- In-distribution (ID) test set includes 1230 proteins belonging to total 230 folds.
- Out-of-distribution (OD) test set includes 204 proteins belonging to 28 folds.

## C DISTRIBUTIONS OVER $\Delta$TM

For the $i^{th}$ protein, we define $\Delta\text{TM}_i = \text{TM}(s_i^{\textbf{Fold2Seq}}, s_i^{\textbf{Native}}) - \text{TM}(s_i^{\textbf{Rosetta}}, s_i^{\textbf{Native}})$. The distributions of $\Delta$TM for the ID and OD tests are shown in Figure 4.

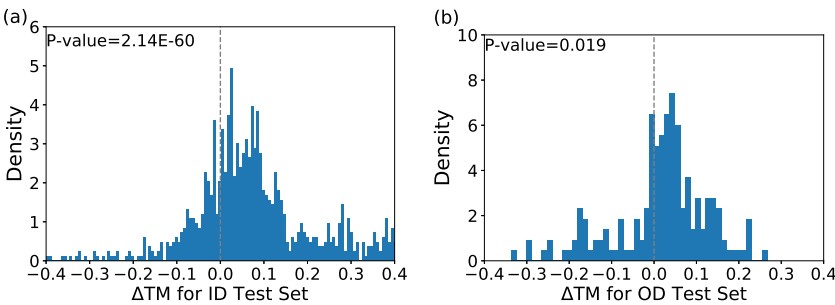

Figure 4: The distributions of $\Delta$TM for two test sets.

## D  COMPARISON OF FOLDED STRUCTURES

In this section, we show some representative folded structures whose sequences are designed by RosettaDesign and Fold2Seq. The folded structures were predicted using iTasser, a state of the art program for protein structure prediction. Figure 5 shows some structures where Fold2Seq performs better than RosettaDesign and Figure 6 shows some structures where RosettaDesign is better.

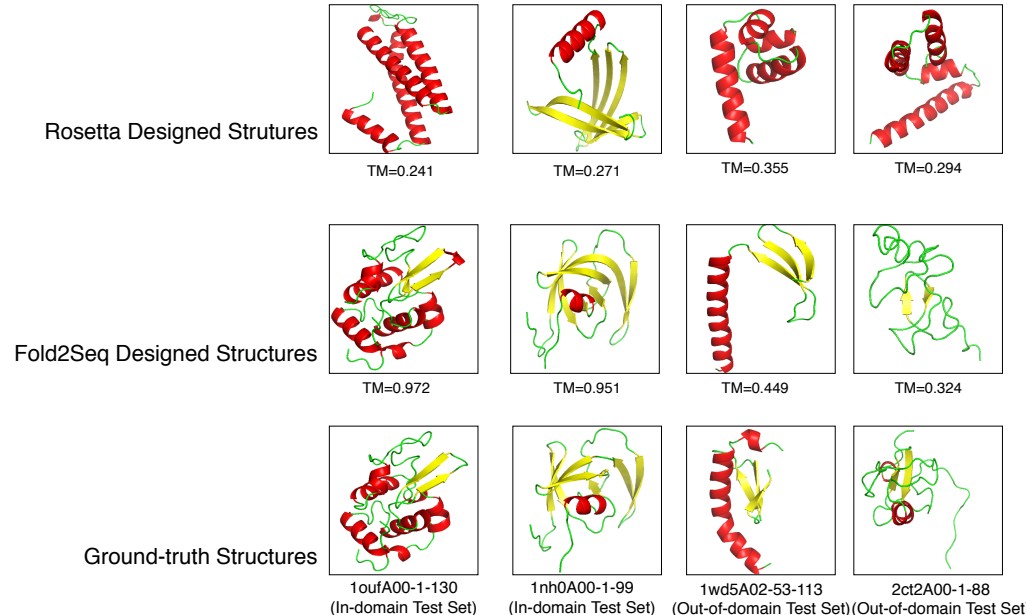

Figure 5: The native and designed structures with $\Delta$TM$_i > 0$. The IDs at the bottom are the CATH domain names of each structure.

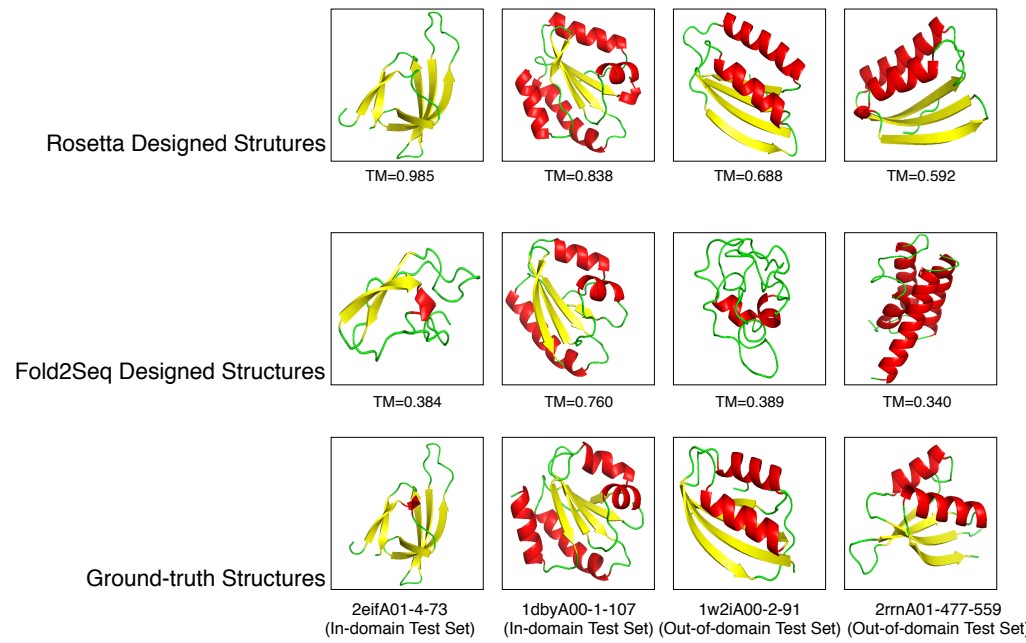

Figure 6: The native and designed structures with $\Delta\text{TM}_i < 0$. The IDs at the bottom are the CATH domain names of each structure.

## E   COMPARISON BETWEEN TWO TRAINING STRATEGIES

In this section, we compare the performance between one-stage training and two-stage training strategies. In the one-stage strategy, we train our model through the 5 loss terms in Eq (4) together. While in the two-stage strategy, we train our model through $L_1$ first and then train it through $L_2$.

We first compare in the learning curves. As the $\textbf{RE}_f$ loss represents the quality of the model, we plot the $\textbf{RE}_f$ loss vs epochs on both training and validation sets.

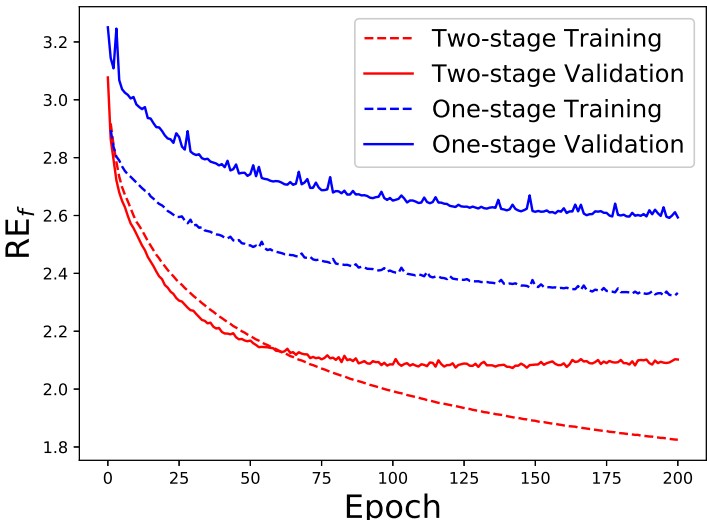

Figure 7: The fold2seq loss($\textbf{RE}_f$) curves of two training strategies on training and validation set.

Table 3: Performance of two training strategies assessed in the sequence domain.

(a) Per-residue Perplexity.

| Model | ID Test | OD Test |
|---|---|---|
| Uniform | 20.0 | 20.0 |
| Natural | 18.0 | 18.0 |
| One-stage strategy | 14.0 | 16.2 |
| Two-stage strategy | **8.1** | **11.9** |

(b) Avg. SeRR $\pm$ std. dev. (%).

| Model | ID Test | OD Test |
|---|---|---|
| Random across two folds | $12.8 \pm 7.94$ | $12.8 \pm 7.94$ |
| One-stage strategy | $19.2 \pm 7.2$ | $16.7 \pm 4.3$ |
| Two-stage strategy | $\mathbf{27.1 \pm 6.31}$ | $\mathbf{24.1 \pm 2.64}$ |
| Random within same fold | $39.1 \pm 9.35$ | $39.1 \pm 9.35$ |

As shown in Fig 7, the two-stage strategy significantly outperformed one-stage strategy. To further demonstrate that, we calculate the per-residue perplexity and the average sequence recovery rate on the two test sets. As shown in Table 3, the same conclusion can be drawn. Those results validate our design choice for two-stage strategy.

## F    TSNE FOR FOLD/STRUCTURE EMBEDDINGS

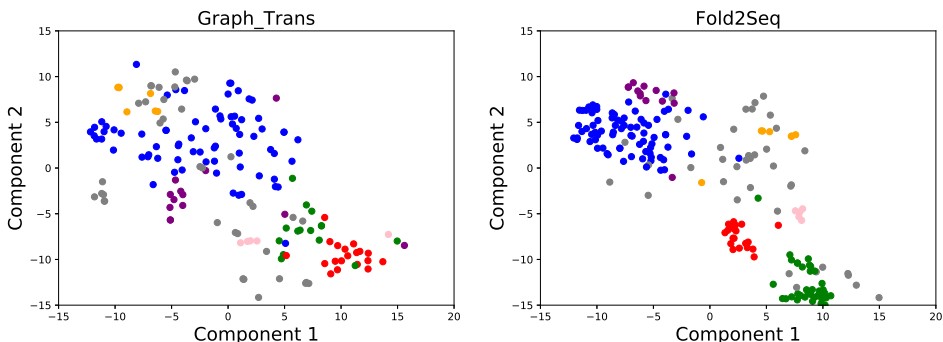

Figure 8: The tSNE visualization of the averaged structure(fold) latent embeddings $h$ after the structure encoder of two methods on OD test set. Each protein is colored by its fold category. Same color indicates the same fold, except that gray points represent outliers, which is defined by its fold having $< 5$ proteins in the test set.

## G    AVG. SEQ RECOVERY RATE PER FOLD.

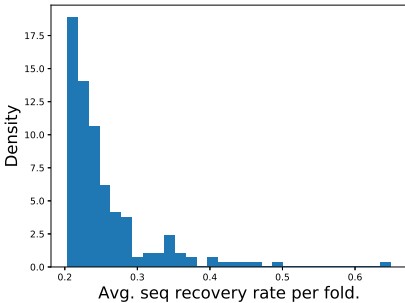

Figure 9: Avg. SeRR per fold for the ID test set.

