# OpenReview forum: "Fold2Seq: A Joint Sequence(1D)-Fold(3D) Embedding-based Generative Model for Protein Design"
_ICLR.cc/2021/Conference — Reject_

### Official Review · AnonReviewer1 · 2020-10-29
**FOLD2SEQ: A JOINT SEQUENCE(1D)-FOLD(3D) EMBEDDING-BASED GENERATIVE MODEL FOR PROTEIN DESIGN**

**Rating:** 6
**Confidence:** 4

**Review:**

This paper tackle the challenge of designing protein sequences that are consistent with a given 3D fold. To address this challenge, the authors propose a transformer-based generative framework that designs protein sequences conditioned on a given fold. There are two central contributions - the first is a novel fold representation, in which the 3D structure is represented by the voxels of secondary structure elements, and then a fold representation is learned via a transformer-based structure encoder. The second is a joint sequence-fold embedding learning framework. The authors use ablation studies to show that learning a joint latent space between sequences and folds enables the model to better capture the the sequence-fold relationship, improving experimental results. The authors provide a good summary of related work.

To proceed with the novel representation, the authors first scale each protein structure to fit into a fixed size cubic box, discretized into fixed size voxels, and extract the alpha carbon coordinates of each amino acid. They assign secondary structure labels drawn from a 4-letter alphabet to each residue, and compute the resulting features of each voxel. These features are used to train a structure encoder, which is used together with a trained sequence encoder to learn a joint sequence-fold embedding. Here fold classification is used for the intra-domain task, while for the cross domain loss they maximize the cosine similarity between the outputs of the sequence and structure encoders in addition to a cyclic loss.

The authors use the data from CATH 4.2 to define two tasks based on a random (ID) split, and a fold-level (OD) split of the data. In each case 2.5% of the data is held out as validation and test sets respectively. They perform various experiments that compare the performance of their approach against cVAE, gcWGAN and RosettaDesign. The performance of this approach is impressive, however I am surprised by the decision to leave out methods that do not focus on the inverse folding problem with a flexible backbone constraint - I would like to see the performance comparison with these methods. Please could the authors adequately justify this decision, or provide these comparisons.

---

### Official Review · AnonReviewer2 · 2020-10-29
**A promising method for protein structure to sequence mapping**

**Rating:** 7
**Confidence:** 5

**Review:**

This is an interesting paper on predicting the underlying amino acid sequence from a given protein tertiary (3D) structure, an important problem with multiple applications. The authors have proposed an elaborate system and performed an impressive set of experiments to demonstrate the efficacy of their model. The proposed method (Fold2Seq) has several novel contributions, which I list below alongside my comments on them.

1. A novel fold representation based on voxels of the density of secondary structure elements (SSEs). This is promising, however, I am not sure how sensitive to small changes in the 3D structure this representation is. The authors shift the structure such that its center of mass is at the origin and its first CA is on the negative side of the Z-axis. What happens only if the first residue is moved around? This is considering that the residues in the N-terminal usually do not have a solid form. Moreover, since the authors focus on short proteins (N < 200), I am curious to know how they handle structures determined by NMR, in which an ensemble of structures is determined, instead of a single structure.
2. A novel joint sequence-fold embedding is learned that is employed in the transformer-based auto-encoder model. This is a great idea and seems to play an important role in Fold2Seq’s performance. Imagine two proteins that have two domains each and share one domain. How would their fold and sequence embeddings look? I am concerned that Fold2Seq might memorize some domains. For example, the drastic difference between Fold2Seq and RosettaDesign TM scores in ID and OD test sets is concerning. While RosettaDesign performs similarly in ID and OD sets, Fold2Seq has a significant drop in performance. I think the distribution of sequence similarities between each test sequence and the closest sequence in ID and OD sets should be provided.
3. I am not sure if splitting the datasets solely based on “fold” is enough to prevent information leakage. I think it should be both based on fold and sequence similarity. Also, since proteins are so small, it might be helpful if their domains are also considered, e.g., using Pfam or computational methods such as doi.org/10.1101/626507.
4. It is not clear to me why the authors did not compare to methods without the flexible backbone constraint (e.g., Ingraham 2019), the rationale is not fully convincing.

---

### Official Review · AnonReviewer3 · 2020-11-04
**An important problem, but this work is too preliminary and makes too many unsubstantiated claims**

**Rating:** 5
**Confidence:** 5

**Review:**

This manuscript presents a method for generating protein sequences conditioned on protein structures. The core idea is to represented protein structures by their secondary structures in 3D space. This voxel grid is then encoded into a vector representation and decoded to a distribution over sequences. The authors propose to learn this model jointly with a sequence encoder, combining the sequence and structure representations to decode the sequence during training. For inference, the sequence encoder component is not used. Learning to generate protein sequence conditioned on structure is an interesting and important problem and has been attracting increasing attention from the ML community. Representing structures as voxel grids is an approach worth exploring and flexible structure representations could be promising. However, it isn’t clear to me that this work achieves those goals and comparisons against key baselines (namely Ingraham 2019) are missing. Furthermore, the authors make many unsupported and unsubstantiated claims about their method. Specific comments and questions follow below.

1.	The claim that this method allows flexible fold representations seems to be undermined by the need to specify a specific structure for which sequences are decoded. How does this method allow for more flexible structure representations than Ingraham 2019 or other methods that require 3D coordinates of each residue? It isn’t clear that this representation is more flexible than that required by RosettaDesign either. The authors need to justify this claim or remove it.
2.	The authors also claim that their method preserves spatial secondary structure information in contrast to others. However, all backbone-based protein structure representations preserve secondary structure information as this is defined by the backbone angles. The authors also claim that their method does not need predefined rules or structures. However, their structure representation is derived from a predefined structure, so this claim makes no sense.
3.	How does the density model loosen the rigidity of structures? The density model only represents a single structure and the authors do not use structure ensembles or other means to represent structural flexibility. I fail to see how this claim is supported by the presented method or experiments. In principle, any structure->sequence model can account for flexibility by ensembling over possible 3D structure.
4.	Have the authors compared their method with Ingraham et al. 2019? Ingraham et al solve the same problem with, arguably, an even more flexible representation of protein structures, because the graph based representation is invariant to rotation and translation of the structure in 3D space. A comparison against Ingraham et al is critical to understand if the proposed voxel-based structure representation is better than other methods. At face value, Ingraham 2019 reports better perplexities with a difficult train/test split and fewer training examples than this work. In fact, Ingraham et al’s out of distribution perplexity surpasses this method’s in distribution perplexity calling into question the utility of this method. The authors should compare their method on the same train/val/test split used by Ingraham et al.
5.	For the in distribution test, how do sequence models fit per fold perform? For example, how well do profile HMMs fit to each fold model the heldout sequences from those folds?
6.	Why are the val and test splits so small (only 2.5%)? Given these splits, how many sequences occur in each?
7.	Because this method encodes structures into a voxel grid, it is not invariant to rotation and translation of proteins in 3D space. The authors align their structures based on center of mass and orient the protein such that the first residue points in the negative direction of the z-axis. How sensitive is the method to misaligned proteins? Is this alignment scheme robust to alternative conformation of the same protein? What happens when structures are rotated? The described alignment scheme also has ambiguity in the rotation around the z-axis. How is this addressed?

Things that would improve my score:
1.	Provide support for unsupported claims in the introduction.
2.	Compare against Ingraham et al 2019.
3.	Examine the sensitivity of this method to rotated structures.

Edit: I have read the authors' response and update  manuscript. I appreciate the new experiments and some of the clarifications. However, some of my fundamental issues with this work are not addressed.

The distinction with Ingraham et al feels forced. It is important to compare head-to-head on Ingraham's dataset in order to truly understand the tradeoffs between these methods and to understand the differences in performance between flexible backbone approaches and a well tuned rigid backbone approach. Why are perplexities for Ingraham et al not reported for all cases? Improving over Ingraham when only low resolution structures or incomplete structures are available is interesting, but I also question how useful this case is. When attempting to do _structure-based design_ of new proteins, how often are fully specified structures not available for those tasks? Ingraham et al presented a variant of their method based only on a flexible structure specification and it isn't clear how that contrasts with this flexible specification. It is certainly possible that this approach is better, but we need a head-to-head comparison to know.

With that in mind, I'll raise my score to 5. I think this work has promise, but is not yet ready for publication in its current state.

---

### Official Review · AnonReviewer5 · 2020-11-06
**Interesting structural representation idea and joint sequence-fold learning, but limiting sequence size**

**Rating:** 6
**Confidence:** 4

**Review:**

The authors of this paper propose a Transformer-based multimodal autoencoder architecture (including a sequence encoder, a structure encoder, and a sequence decoder) for protein design. In the method, a claimed novelty is that a 3D proteins structure is represented by 3D voxels of density of secondary structures. Transformer is used for sequence modelling and structural modelling. A another major claim of this paper is the joint sequence-fold representation learning. The objective function reflects the needs for both intra-domain representation learning and cross-domain representation learning. Comparisons with a principled-based method - RosettaDesign and two machine learning methods - cVAE and gcWGAN at both sequence level and structural level show that the proposed method can achieve better performance on sets of proteins with limited sizes. Overall, this is an interesting work for 3D structure learning and design. However, I also have the following concerns.

Majors:
1. A possible major drawback of the proposed structural representation method is that it relies on rescaling of very large structures to the 3D frame of fixed size. In the experiments, proteins with more than 200 amino acids were filtered out. Moreover, in the inference phase, this representation can cause a complication to the length of generated sequences.

2. In the training phase, it is unclear whether L_1 and L_2 are optimized alternatingly along multiple epochs, or L_1 is optimized for multiple epochs then L_2 for multiple epochs. How much gain was obtained in comparison with just direct optimization of the objective function with five terms in terms of learning curves and performance at sequence and structure levels? I guess the unbalanced training progress is due to the fact that training sequences is easier than training 3D structures.

Minors:
position(3D -> position (3D
transformer -> Transformer
In the text, the authors state that "we consider to maximize the cosine similarity (shown as the ‘Cosine Similarity’
pink block in Fig 2) ...", however, "Cross Similarity" is given in Figure 2.
a MLP -? an MLP
Pytorch -> PyTorch
ai-driven -> AI-driven [in the references, check similar issues]

---

### Decision · Program_Chairs · 2021-01-07
**Final Decision**

**Decision:**

Reject

**Comment:**

This is a promising idea, without enough empirics to substantiate its potential utility, and also with a lack of clarity on the importance of the outlined task itself (fold-based rather than structure-based conditional sequence generation). There remain concerns about the lack of a more comprehensive comparison to methods for structure-to-sequence (e.g. Ingraham was added during the revision but only in a limited capacity), or easy generalizations of them, and about the quality of some of the presented results. Additionally, the concern about sensitivity to rotational invariances, and related issues wrt the fixed-size cubic grid were not satisfactorily addressed. As a side-note, the quality of the manuscript in terms of scholarliness of presentation was overall lacking.